# Tolosa–Hunt Syndrome and Hemorrhagic Encephalitis Presenting in a Patient after COVID-19 Vaccination Followed by COVID-19 Infection

**DOI:** 10.3390/brainsci12070902

**Published:** 2022-07-10

**Authors:** Anca Elena Gogu, Andrei Gheorghe Motoc, Any Docu Axelerad, Alina Zorina Stroe, Andreea Alexandra Gogu, Dragos Catalin Jianu

**Affiliations:** 1Department of Neurology, “Victor Babes” University of Medicine and Pharmacy, 300041 Timisoara, Romania; anca.gogu@umft.ro (A.E.G.); dcjianu@yahoo.com (D.C.J.); 2Centre for Cognitive Research in Neuropsychiatric Pathology (Neuropsy-Cog), Faculty of Medicine, “Victor Babeș” University of Medicine and Pharmacy, 300041 Timișoara, Romania; 3Department of Anatomy and Embryology, “Victor Babes” University of Medicine and Pharmacy, 300041 Timisoara, Romania; amotoc@umft.ro; 4Department of Neurology, General Medicine Faculty, “Ovidius” University, 900470 Constanta, Romania; docuaxi@yahoo.com; 5“Victor Babes”, Medicine Faculty, University of Medicine and Pharmacy, 300041 Timisoara, Romania; agogu3@yahoo.com

**Keywords:** COVID-19 infection, Tolosa–Hunt syndrome, hemorrhagic encephalitis, infectious/autoimmune vasculitis

## Abstract

The presence of neurological symptoms within the clinical range of COVID-19 disease infection has increased. This paper presents the situation of a 45-year-old man having the medical antecedent diabetes mellitus, who presented to the emergency department with fever, headache, and respiratory symptoms, nine days following vaccination with the Ad26.COV2-S COVID-19 vaccine. The patient tested positive for SARS-CoV-2 based on nasal polymerase chain reaction (RT-PCR). Two weeks after the presentation, he developed Tolosa–Hunt Syndrome, an autoimmune phenomenon, with painful left ophthalmoplegia. Significant improvement was seen in terms of his discomfort; however, ptosis and ocular mobility improved only moderately after treatment with intravenous methylprednisolone, and the patient was discharged on a new insulin regimen. The patient returned after four weeks and the neurological exam results showed significant signs of right hemiparesis, mixed aphasia, incomplete left ophthalmoplegia, severe headache, and agitation; after a few days, the patient experienced a depressed level of consciousness and coma. The patient’s clinical condition worsened and, unfortunately, he died. MRI brain images revealed multiple ischemic strokes, meningitis, infectious vasculitis, and hemorrhagic encephalitis, which are all serious complications of COVID-19.

## 1. Introduction

Tolosa–Hunt syndrome is a painful illness characterized by idiopathic, self-limited inflammation affecting the cavernous sinus, and is often responsive to corticosteroid treatment [1]. This autoimmune phenomenon has been associated with numerous reported triggers, including COVID-19 vaccination or infection with COVID-19 [2,3]. The antibodies originate from the immunological system of the host to oppose the cross-reacted virus and bind to the cranial/peripheral nerves, causing neuronal impairments [4,5].

Hemorrhagic encephalitis is another serious complication of COVID-19 infection. In critically ill patients, SARS-CoV-2 infection has been linked to coagulopathies including thrombocytopenia, increased D-dimer, and delayed prothrombin time, which may lead to hemorrhaging [6]. SARS- CoV-2-induced ACE2 downregulation may result in vasoconstriction and blood pressure spikes, which may trigger arterial wall rupture and hemorrhaging. Hypoxemia, systemic inflammation, and direct viral invasion all contribute to the development of this disease.

This article describes a case of Tolosa–Hunt syndrome that presented twenty-three days subsequent to vaccination with the Ad.26.CoV 2-5 vaccine, and fourteen days after COVID-19 infection. A review of the existing literature is also presented. After a further four weeks, the patient, who had improperly managed diabetes mellitus, showed significant neurological involvement comprising repeated ischemic strokes, meningitis, infectious vasculitis, and fatal hemorrhagic encephalitis.

This research was authorized by the Ethics Committee for Clinical Studies at the Emergency Clinical Hospital of the County of Timisoara (registration number 307/20.06.2022) and was undertaken according to the Declaration of Helsinki. The individual provided written informed permission for enrolment.

## 2. Case Presentation

The patient was a 45-year-old man, previously known to have diabetes mellitus, who presented to the emergency department nine days after receiving COVID-19 vaccination (Ad 26. COV2-5). The injectable vaccination administered to the deltoid was COVID-19 Vaccine Janssen (Ad.26. COV2-S) manufactured by Johnson & Johnson. The active ingredient is a type 26 adenovirus, which encodes the spike glycoprotein of SARS-CoV-2. Using recombinant DNA technology, it is produced in Cell Line PER C6 Tet R. COVID-19 protection is provided by the vaccination, which stimulates the immune system to create antibodies and activates specialized T cells that work against the virus. None of this vaccine’s components can induce COVID-19 infection. The symptomatology of the patient at presentation included fever, headache, and moderate respiratory symptoms. The diagnose of COVID-19 infection was obtained by nasal polymerase chain reaction (RT-PCR), and the patient was treated in the internal medicine department. His chest-computed tomography (chest-CT) revealed 25% lung damage with bilateral multifocal infiltrates. Two weeks after the presentation, he developed a severe left-side headache, periorbital pain on the same side with progressive left upper eyelid ptosis, decreased visual acuity to blindness in the left eye, and left periorbital cellulite. Furthermore, the patient presented extraocular motility deficits with left cranial oculomotor nerves palsies (cranial nerve III, trochlear nerve IV, and abducens nerve VI), an afferent pupillary defect, and stereotyped episodes of intensely painful electric-like shocks along the ipsilateral 1st and 2nd branches of the trigeminal nerve. The remainder of his neurological exam was normal. He was transferred to our hospital for additional treatment and neurological management.

When arriving at the Department of Neurology, the patients’ vitals were stable, and the neurological examination has conducted, as previously described. The modified paraclinical tests showed WBC of 12,540/μL, serum glucose of 338 mg/dL, and HbA1c of 12.2%; serum inflammatory workup showed elevated hsCRP of 19.8 mg/L, fibrinogen of 638 mg/dL, D-dimer of 54 ng/mL, and VSH of 44 mm/h. The thyroid hormones levels were FT3 of 3.71 pmol/L, FT4 of 14.5 pmol/L, and TSH of 1.02 mUI/L. The Fluorescent Treponemal Antibody Test and HIV Combo (Antibody Anti-HIV + Antigene HIV p24 = 0.06 Index) were negative. The hepatitis panel and Lyme antibodies were also negative. CSF indicated high glucose levels of 137 mg/dL (normal 40–70 mg/dL) and normal protein of 0.62 g/L, without white and red blood cells when a lumbar puncture was conducted. CSF studies included culture and Gram staining, which were negative for the meningitis-encephalitis panel. The increased CSF glucose was interpreted as a normal effect of the initiation of i.v. Methylprednisolone 1 g/day, for three days, and because the patient had diabetes mellitus. SARS-CoV-2 by RT-PCR was positive and SARS-CoV-2 IgG antibodies (antiSpike S1) were increased to more than 200 BAU/mL.

Contrast-enhanced head-computed tomography (head-CT) was conducted upon hospital admission and demonstrated hyperattenuation of the left cavernous sinus and left lateral sinus, which was concerning for sinus thrombosis. MRI of the brain and orbits with and without contrast, in addition to magnetic resonance angiography (MRA)/magnetic resonance venography (MRV) of the head, showed an inflammatory process in the areas of the left cavernous sinus and orbital apex with perineural enhancement surrounding the left optic nerve sheaths. The presence of thrombus in the cavernous sinus was refuted by cerebral MRA/MRV. No vascular malformation of the left internal carotid artery was found. The MRI findings were suggestive of Tolosa–Hunt syndrome (Figure 1).

Treatment was administered with one gram of i.v. Methylprednisolone for three days, after which, the patient was switched to prednisone orally with a slow taper. He showed significant improvement in terms of pain, with minimal improvement in eye motility and visual acuity. The patient was released on a new insulin regimen.

The patient returned after one month to the emergency department with right hemiparesis, mixed aphasia, incomplete left ophthalmoplegia, severe headache, and agitation, without fever. He was tested again for SARS-CoV-2 with nasal polymerase chain reaction (RT-PCR) and was found to be positive. His vitals were stable and non-contrast brain CT showed left capsulo-lenticular and temporal hypodensities. MRI brain images with and without contrast showed aspects in T2, Flair, and DWI hypersignal without contrast capture of the left temporal cortex, respectively, the hippocampal and ipsilateral parahippocampal region. An MRI brain scan also revealed multiple images in hypersignal T2, Flair, and DWI with dimensions of 2–10 mm located juxtacortically and deeply periventricular fronto-temporo-parietal bilateral. The appearance was suggestive of encephalitis (Figure 2).

The patient’s laboratory results showed WBC of 9440 cells/μL, blood glucose of 297 mg/dL, HbA1c of 10.7%, hsCRP of 16.7 mg/L, fibrinogen of 475 mg/dL, D-dimer of 81 ng/mL, and ferritin of 169 μg/L. Thyroid hormones were normal, and HIV screening and liver and renal function were unremarkable. ANTISARS-CoV-2 IgG-antiSpikeS1 was >200 BAU/mL and total ANTISARS-CoV-2 (IgA IgM IgG) was 835 S/CO. A repeat lumbar puncture was undertaken, which showed elevated glucose of 104 mg/dL, normal protein of 0.62 g/L, 625 red blood cells, and no white blood cells. CSF bacteria and fungi showed no growth. The patient was started on human immunoglobulins due to concerns regarding autoimmune/inflammatory encephalitis, which can cause similar presentations including ischemic or hemorrhagic strokes, meningitis, and encephalopathy.

The patient continued to have worsening encephalopathy, and, after seven days, he presented with a depressed level of consciousness and extreme agitation. A repeat MRI brain scan with and without contrast showed extension of the lesion in the left fronto-parieto-temporal lobes with hypersignal aspects on the T2, Flair, and DWI images. Small hemorrhagic lesions were also shown in the left temporal lobe (hyperintense on T2 images) and in the posterior limb of the internal capsule. A hyperdense aspect capturing the contrast of the left middle cerebral artery wall was highlighted. The appearance was suggestive of the infectious/autoimmune vasculitis. The fronto-parieto-temporal meninges showed contrast capture on the Flair images (Figure 3).

On day eight, the patient became comatose with altered neurological status and was intubated, with mechanical ventilation in the intensive care unit. He required i.v. fluids, pressors, ceftriaxone, vancomycin, and antifungals due to concerns regarding sepsis. A CT of the brain was obtained in light of the acute hemorrhage of intraparenchymal fronto-parieto-temporal lobes, showing important edema (Figure 4).

A neurosurgeon was consulted but the patient’s clinical condition worsened and he was not a candidate for surgery.

The laboratory results were notable for WBC of 11,571/μL, blood glucose of 260 mg/dL, hsCRP of 235.1 mg/L, fibrinogen of 1545 mg/dL, D-dimer of 378 ng/mL, prothrombin time of 15.1 s., APTT of 34.7 s., and INR of 1.37. The patient’s clinical condition continued to deteriorate and, unfortunately, after two weeks, he died.

## 3. Discussion

Increasing amounts of scientific evidence indicate that severe acute respiratory syndrome coronavirus-2 (SARS-CoV-2) potentially affects the brain [7]. Diagnostic considerations in painful ophthalmoplegia (Tolosa–Hunt syndrome) must include both COVID-19 infection and its vaccination to better clarify possible post-infectious or autoimmune disorders [3]. Recently, Chuang et al. described a case after immunization with mRNA COVID-19 and Etheridge et al. described another case of Tolosa–Hunt syndrome subsequent to COVID-19 infection [2,3]. Our patient, who presented nine days after receiving the COVID-19 vaccination (Ad.26.CoV2-S), was found to be positive for SARS-CoV-2 based on nasal polymerase chain reaction (RT-PCR); two weeks later he developed Tolosa–Hunt syndrome. This raises the question of what caused the symptoms. Was it an adverse reaction to the vaccine or was it associated with SARS-CoV-2 infection? Most likely, a dual mechanism occurred due to COVID-19-related immune dysregulation and association with an infectious disease. 

Under normal circumstances, the endothelium may react to a variety of hemodynamic and humoral stimuli by releasing mediators that regulate vascular tone, cellular adhesion, coagulation, and vessel wall inflammation. The endothelium plays a crucial function in regulating the immune response, since it controls leucocyte migration into extravascular areas and protects against infections. Endothelium dysfunction is the mechanism of COVID-19 and the underlying cause of the majority of clinical signs of SARS-CoV-2 infection [8].

Autoimmune neurological phenomena have been associated with influenza, mumps, and rubella vaccinations [9]. COVID-19 immunization has shown important advantages throughout this pandemic, despite the observed adverse effects.

From our perspective, SARS-CoV-2 infection is not a result of the administered vaccination, but rather a consequence of a series of events. It is likely, however, that the vaccination caused an immunological cascade and that the SARS-CoV-2 infection occurred within a treatment “window” in which immunity was suppressed and Anti-SARS-CoV-2 antibodies had not yet been produced.

A substantial number of patients having a COVID-19 infection had acute cerebrovascular events, including ischemic stroke [4]. The general incidence of encephalitis in COVID-19 patients is less than 1 percent, but may reach as high as 6 to 7 percent in patients with severe illness [10]. The clinical symptoms, such as a low state of consciousness, seizures, headache, and motor impairments, are comparable to those of other types of encephalitis; however, the mortality rate was determined to be around 13 percent, which is considerably higher compared with other types of encephalitis [11]. Encephalitis may present with meningeal and parenchymal symptoms, white matter inflammation, vasculitis, and, in rare cases, hemorrhagic encephalitis [4,12]. The mechanism of hemorrhagic encephalitis secondary to COVID-19 is multifactorial but incompletely known. It is hypothesized that hemorrhagic encephalitis in COVID-19 infection could occur consequent to direct viral invasion of the CNS, and the intense inflammatory response caused by SARS-CoV-2 infection that may induce blood–brain barrier (BBB) breakdown [13]. An enhanced BBB permeability, which may enable the circulation of peripheral cytokines through the CNS and provoke an indirect neuroinflammatory reaction, may be accountable for hemorrhagic encephalitis [13,14]. Considering that the endothelium also expresses ACE2 receptors, the latter may be activated by cytokine release in addition to viral damage. The depletion of ACE2 by SARS-CoV-2 may cause the imbalance of the renin angiotensin system, which may result in endothelial dysfunction [4,15]. SARS-CoV-2-induced ACE2 downregulation can result in vasoconstriction and blood pressure spikes that can cause arterial wall rupture and hemorrhage. Other possible mechanisms include hypoxia, toxicity, and metabolic and electrolyte imbalances, which are associated with cerebral microhemorrhages in the subcortical areas. COVID-19-related coagulopathy is involved in the etiology of cerebral microhemorrhages. There are two separate mechanisms: coagulation irregularities represented by an increase in procoagulant factors (fibrinogen, platelets, D-dimer) and an inflammatory response with increased levels of highly sensitive C-reactive protein (hsCRP) and erythrocyte sedimentation rate (ESR).

Given the evolution of encephalitis in our patient with cerebral microhemorrhages followed by a massive hemorrhage, a possible explanation may be vasculitis, which affected the large cerebral arteries. This was shown in a cerebral MRI with contrast.

## 4. Conclusions

To the best of our knowledge, this is the first report of a case of Tolosa–Hunt syndrome presenting after COVID-19 vaccination and subsequent infection with SARS-CoV-2, followed by hemorrhagic encephalitis. Tolosa–Hunt syndrome and hemorrhagic encephalitis are rare but serious complications of COVID-19 infection. The etiology of these diseases is poorly understood and further research in the field represents an ongoing challenge for neurologists.

## Figures and Tables

**Figure 1 brainsci-12-00902-f001:**
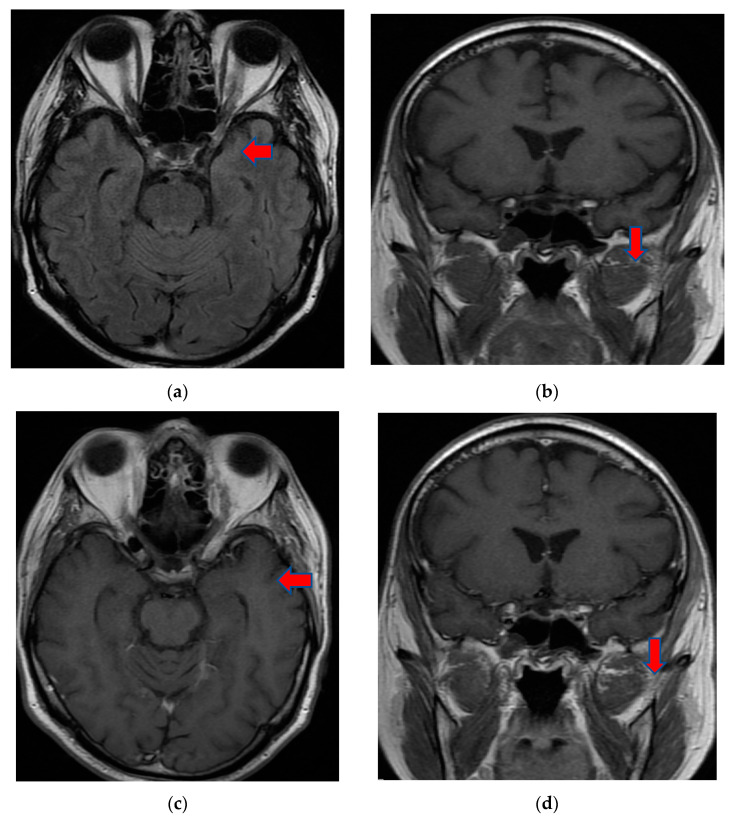
MRI T2 axial Flair (**a**) and coronal T1 FSE (**b**) images demonstrated perineural tissue extending into the left cavernous sinus. Postcontrast T1 axial FSE (**c**) and coronal T1 FSE (**d**) images showed an inflammatory process involving the left cavernous sinus and orbital apex with perineural enhancement surrounding the left optic nerve sheath. MRI: magnetic resonance imaging; Flair: fluid attenuated inversion recovery; FSE: fast spin-echo.

**Figure 2 brainsci-12-00902-f002:**
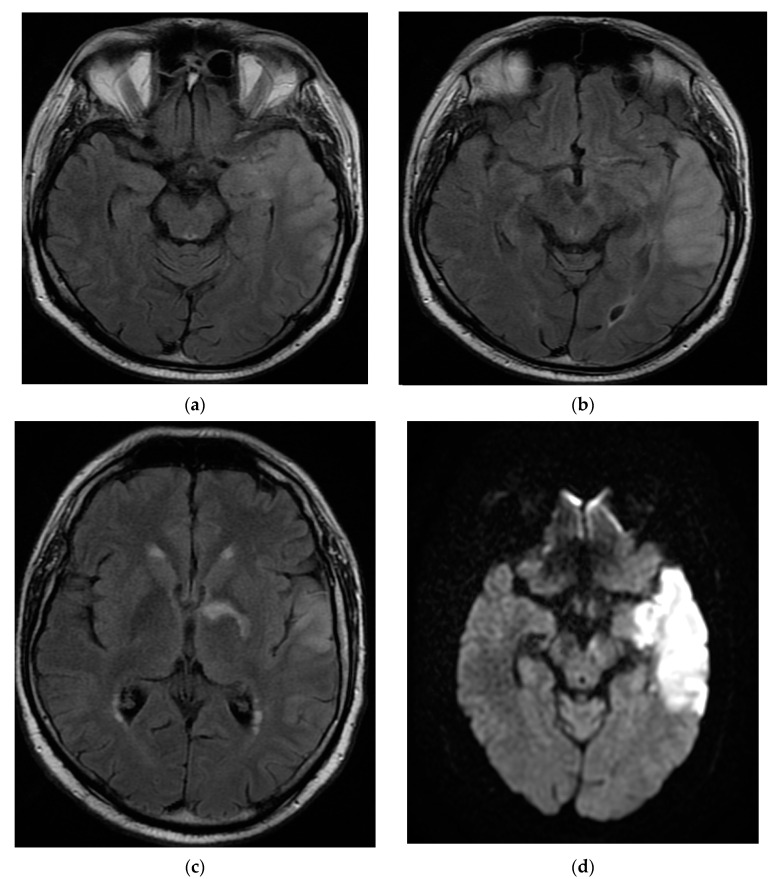
MRI brain images with contrast showed aspects of T2 axial Flair hypersignal in the left temporal lobe (**a**,**b**), respectively, in the hippocampal and parahippocampal region (**c**). The lesion is hyperintense on the DWI axial image (**d**). MRI: magnetic resonance imaging; Flair: fluid attenuated inversion recovery; DWI: diffusion weighted imaging.

**Figure 3 brainsci-12-00902-f003:**
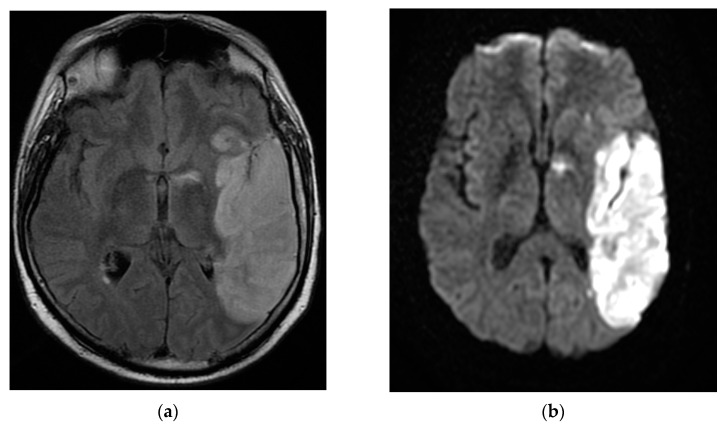
MRI T2 axial Flair (**a**) and axial DWI (**b**) images demonstrated extension of the lesion in the fronto-parieto-temporal left lobes. Coronal T1 FSE (**c**) and axial T1 FSE (**d**) with contrast showing a hyperdense aspect of the left middle cerebral artery wall, suggestive of infectious/autoimmune vasculitis.

**Figure 4 brainsci-12-00902-f004:**
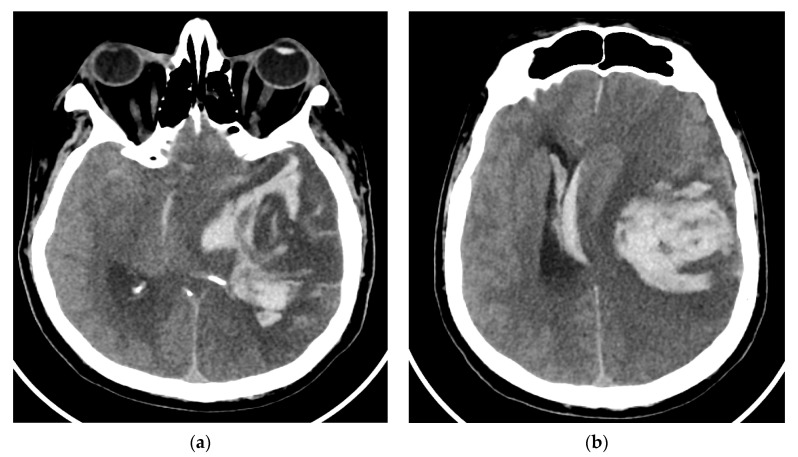
(**a**,**b**), Brain-CT showed acute hemorrhage of intraparenchymal fronto-parieto-temporal lobes with important edema and ventricular extension.

## Data Availability

Third-party data restrictions apply to the availability of these data. The data were obtained from Timisoara County Emergency Clinical Hospital and are available from the authors with the permission of the Institutional Ethics Committee of Clinical Studies of the Timisoara County Emergency Clinical Hospital.

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
