# Peer review of "Tolosa–Hunt Syndrome and Hemorrhagic Encephalitis Presenting in a Patient after COVID-19 Vaccination Followed by COVID-19 Infection"

_brainsci, 2022, doi:10.3390/brainsci12070902_

Round 1

Reviewer 1 Report

The story of Ad.26.CoV 2-5 vaccine continues to raise important warnings. The US Food and Drug Administration paused its use in April 2021 due to the reported concern of clotting disturbances. The pause was lifted 10 days later, but the concern remains a fluid one as your case report indicates it.

Author Response

Dear reviewer,

We are grateful for your help and guidance.

We have added valuable information about the physiological mechanism that can be involved in related cases.

Reviewer 2 Report

Dr. Gogu et al. present a case of Tolosa-Hunt syndrome and hemorrhagic encephalitis that followed COVID-19 vaccination and subsequent COVID-19 infection. The manuscript is well-written and the case is well-presented. The discussion and references are adequate.

It is of course preferable to have a tissue examination performed, either in the form of biopsy or autopsy, to elucidate the actual pathology underlying the clinical presentation. For instance, one of the cases presented in reference #9 had invasive fungal infection. On the other hand, it is understandable that obtaining tissue is not always possible; therefore, the presentation remains as a clinicoradiologic one.

Although the vaccine code# is provided, it is not clear to me what type of vaccine was administered. The company and mechanism information will be useful to possibly present more detail. 

In addition, it will also be useful to provide some discussion on the general basics and statistics of getting COVID infection despite being vaccinated. I think this discussion also provides a good opportunity to provide some information on the potential side effects of COVID vaccines. These can then provide some insight into the potential issues and possible connections (or absence thereof) with this clinical presentation.

Author Response

Dear reviewer,

We are grateful for your help and guidance.

We have added valuable information about the physiological mechanism that can be involved in related cases.

An autopsy was not possible, as the family did not agree. It would have been great to have the opportunity to examine the brain biopsy. Of course, we also though of a fungal infection of the brain but the blood cultures were negative.

Lines 117-123

The deltoid intramuscular vaccine administrated was COVID-19 Vaccine Janssen (Ad.26. COV2-S) by Johnson & Johnson. The active substance is adenovirus type 26 which encodes the spike glycoprotein of SARS-COV-2. It is produces in a Cell Line PER C6 Tet R by recombinant DNA technology. The vaccine stimulates the immune system to produce antibodies and activates specialized T lymphocytes, acting against the virus, providing protection against COVID-19. None of the components of this vaccine can cause COVID-19 disease.

Reviewer 3 Report

This paper is fascinating: a really rare case report
The introduction describes well the topic
The case report is exciting and also correlated by a good figure and MRI imaging
I have only one suggestion: I think you should explain better what you hypothesize a relation between covid-19 vaccines and covid infection
The covid-19 is well demonstrated that is a sort of systemic disease with a diffuse endothelial damage
Please explain better your idea about a possible physio-pathological mechanism
conclusion is ok

Author Response

Dear reviewer,

We are grateful for your help and guidance.

We have added valuable information about the physiological mechanism that can be involved in related cases.

Lines 452-456

In our perspective, SARS-Cov-2 infection is not a result of the administered vaccination, but rather a consequence of a series of events. It is likely, however, that the vaccination caused an immunological cascade and that the SARS-Cov-2 infection occurred within a treatment "window" in which immunity was suppressed and Anti-SARS-Cov-2 antibodies had not yet been produced.

Lines 428-434

Under normal circumstances, the endothelium may react to a variety of hemodynamic and humoral stimuli by releasing mediators that regulate vascular tone, cellular adhesion, coagulation, and vessel wall inflammation. Endothelium plays a crucial function in regulating the immune response, since it controls leucocyte migration into extravascular areas and protects against infections. Endothelium dysfunction is the mechanism of COVID-19 and the underlying cause of the majority of clinical signs of SARS-Cov-2 infection [8].
